# Urban surface water flood modelling – a comprehensive review of current models and future challenges

Kaihua Guo[1], Mingfu Guan[1*], Dapeng Yu[2]

[1]Department of Civil Engineering, the University of Hong Kong, Hong Kong, China
[2]Geography and Environment, Loughborough University, Loughborough, LE11 3TT, United Kingdom

[*] *Correspondence to*: Mingfu Guan (mfguan@hku.hk)

**Abstract.** Urbanisation is an irreversible trend as a result of social and economic development. Urban areas, with high concentration of population, key infrastructure, and businesses are extremely vulnerable to flooding and may suffer severe socio-economic losses due to climate change. Urban flood modelling tools are in demand to predict surface water inundation
caused by intense rainfall and to manage associated flood risks in urban areas. These tools have been rapidly developing in recent decades. In this study, we present a comprehensive review of the advanced urban flood models and emerging approaches for predicting urban surface water flooding driven by intense rainfall. The study explores the advantages and limitations of existing model types, highlights the most recent advances and identifies major challenges. Issues of model complexities, scale effects, and computational efficiency are also analysed. The results will inform scientists, engineers, and decision-makers of
the latest developments and guide the model selection based on desired objectives.

## 1 Introduction

Flooding is a common, widespread and frequent natural hazard that causes severe socio-economic loss and environmental impact worldwide (Barredo, 2009; Teng et al., 2017). Flood risk is exceptionally high in urban areas where the land surface varies, and anthropogenic activities cause remarkable changes in hydrological processes (Guan et al., 2015; Sillanpää et al.,
2015). Urban surface water flooding (also called pluvial flooding) is generally triggered by intense rainfalls when the capacity of urban drainage systems is overwhelmed (Falconer et al., 2009; Chen et al., 2015). Flood risk management has historically focused on fluvial and coastal flooding, with significantly less emphasis on urban surface water flooding. Although often associated with shallow water, unlike fluvial and coastal flooding, the impact of surface water floods can be equally widespread. For example, the Pitt Review (2008) commented that during floods that affected the UK in the summer of 2007,
two thirds of the damage in urban areas was caused by surface water flooding, for which no models, forecasts, warnings or management strategies existed. In cities, different from the rural area, where impervious surfaces make surface water flooding most likely, the impacts can be particularly severe. Direct damage, via inundation of properties and critical infrastructure (e.g. electricity sub-stations, bridges and drainage system), and indirect consequences, such as loss of productivity and business opportunities, can occur (Barredo et al., 2012). Numerous studies have reported that urban surface water flooding has caused

the tremendous socio-economic loss, which is expected to increase in severity and frequency in the future with urbanisation, economic development, and more frequent extreme weather (UNISDR, 2015; IPCC, 2013; Bernet et al., 2017; Barredo, 2009; Zhou et al., 2013; Moncoulon et al., 2016). IPCC (2014) indicated that climate change will cause extreme precipitation events more intense and frequent in many regions, so leading to greater flood risks. Therefore, it is crucial for effective flood risk management to develop modelling techniques that simulate and predict the dynamic processes of storm-induced urban flooding.

Fluvial and coastal flood modelling and inundation mapping have been studied extensively and have become common practices in past decades (Neelz et al., 2013; Rubinato et al., 2019). These models greatly rely on the quality of topographic data and hydrological data as input. Different from river- and rural-catchment terrain, urban areas generally have more complex and irregular topography with buildings, drainage networks and other critical infrastructures. The dynamics and non-linear interactions of hydrological, hydrodynamic and hydro-morphological processes in such topography present substantial modelling challenges. Also, the heterogeneity of urban surface challenges the parametrisation of urban flood models. Significant efforts have been made to advance the development of a reliable model, which is essential for building urban flood resilience (Song et al., 2014). Owing to the progress in computational power and data availability, the quality and complexity of these models have steadily increased, beginning with a simplified model framework a decade ago to more sophisticated numerical models in recent years (Mignota et al., 2019). Amongst these, four groups of models are the most concerned in the ongoing urban flood simulation research (Figure 1): (1) drainage network models (e.g. Djordjevic et al., 1999; Simoes et al., 2010); (2) shallow water-based models that solve simplified or full shallow water equations (SWEs) with a varying conceptualisation of sewer flows, e.g. the diffusive version of LISFLOOD-FP (Bates et al., 2010), CityCAT (Glenis et al., 2018) and HiPIMS (Xia et al., 2019); (3) hydrogeomorphic approaches that predict the inundation area based on geomorphic features (e.g. Nardi et al., 2013; Di Baldassarre et al., 2020); and (4) other methods such as cellular automata (CA) and artificial neural networks (ANN) (Ghimire et al., 2013; Guidolin et al., 2016; Berkhahn et al. 2019).

The accuracy and efficiency are generally two key indicators for evaluating the performance of these models but are often in conflict with each other. For the fast-simplified models, some key information is lost, leading to less accuracy in the results, without even including the key inundation feature. Sophisticated models can predict more flood information but at the cost of more high-quality data input and expensive computation. Although still in its infancy, models for reproducing the interaction of surface waters and drainage flows are being developed (Leandro et al., 2009; Seyoum et al., 2011; Bazin et al., 2014). Large-scale urban flood simulation is still a challenge for these models due to the requirement of extensive input data, such as a pipe network. Several studies have reviewed flood models for fluvial and coastal inundation (e.g. Teng et al., 2017), but none has systematically reviewed the specific type of models for urban surface water flooding driven by intense rainfall. This paper aims at evaluating the available urban flood models by exploring their advantages and disadvantages for various applications.

The health risk associated with pollutant transport during urban flooding is also an important issue to be modelled and overcome. Urban flooding can cause a surcharge of sewer flow, flush pollutants and wastewater to public area, so causing health risks for the people, such as breakout of epidemic disease, and drinking water pollution (Beg et al., 2020). However, surface water pollution heavily relies on the surface water dynamics. Therefore, this review only focusses on flood hydrodynamic modelling. The paper will first present the systematic methodology for selection of literature in Section 2. Then, the methods of different types of urban flood models in the literature are summarised and overall described in Section 3. In Section 4, the main advantages and limitations of each model type are evaluated and discussed, which provides guidance and suggestions for the optimization of the method/model in practical. Section 5 discusses future research needs and challenges. Conclusions are drawn in Section 5.

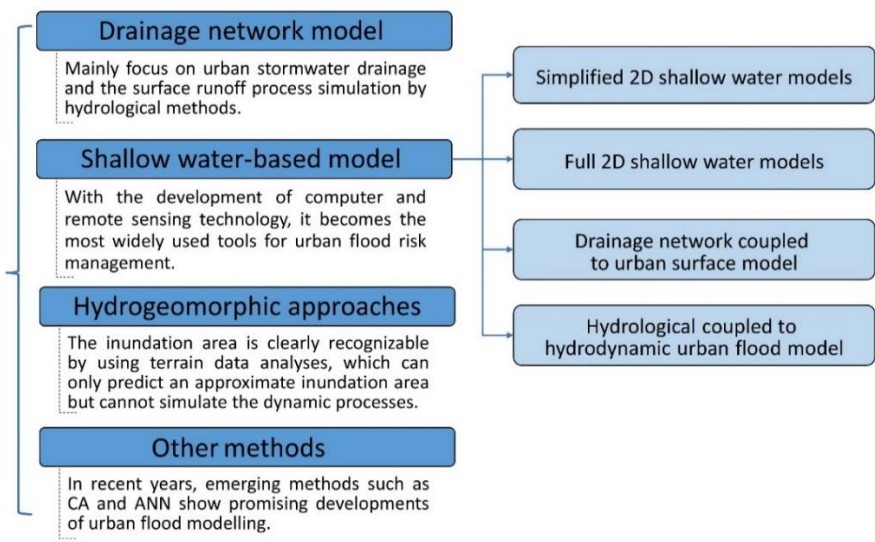

**Figure 1: Four main groups of models in the ongoing research.**

## 2 Methodology

In this study, two steps were taken with pre-defined criteria to systematically filter and select all potential papers based on an established research methodology (Booth et al., 2011). The following is a detailed description of how to use this method for a systematic overview (Gradeci et al. 2019).

The research theme in the present study was determined as how urban surface flood models have been developed and applied. In order to establish a research entry point and the preliminary investigation scope, the first step was to identify the keywords as search terms. With reference to the selected representative literature (Djordjevic et al., 1999; Simoes et al., 2010; Bradbrook et al., 2004; Bates et al., 2010; Glenis et al., 2018; Xia et al., 2019; Ghimire et al., 2013; Guidolin et al., 2016; Berkhahn et al. 2019), the keywords are determined as showing in Table 1. The review was conducted in the HKU library (Engineering) (the

core databases are Engineering village 2, INSPEC, SciTech premium collection, Scopus and Web of Science) and Google Scholar. Furthermore, this study concentrated on research models, rather than commercial models, which typically involve software applications.

The second step is the final screening, which is based on the chosen electronic database (HKU library (Engineering)) of peer-reviewed literature. The pre-defined exclusion criteria were set to screen the literature (Table 2). The search scheme is illustrated in Figure 2 and outlining the number of papers retained at each stage. There was no limitation to the research period and the search conducted on 06 October 2020. For the same model in different literature sources, newly published papers are preferred. And journal papers take precedence over conference papers. Finally, 48 papers were selected. Literature pool is listed in the supplementary table.

**Table 1: review keywords.**

| What | Where | How |
|------|-------|-----|
| • Urban<br>• Flood/inundation<br>• Numerical model | • Storm<br>• Precipitation<br>• Rain<br>• Surface water<br>• Pluvial flood | • Analysis<br>• Assessment<br>• Risk<br>• Validation<br>• Verification |

**Table 2: Exclusion criteria.**

| No. | Exclusion criterion | What is excluded |
|-----|---------------------|------------------|
| 1 | Qualitatively based on the type of literature | Literature other than journal articles and conference papers printed in English; Proceeding Papers; Books/Chapters |
| 2 | Scientifically based on keywords, titles | Focus on other types of floods like fluvial flood or coastal flood; urban land use pattern analysis; description of urban flood risk; flood risk assessment; large-scale flood research. |
| 3 | Scientifically based on abstract | |
| 4 | Scientifically based on article, quality assessment | |
| 5 | Scientifically based on the model updating | Algorithms or frameworks have been improved and published, only focus on applications of published model |

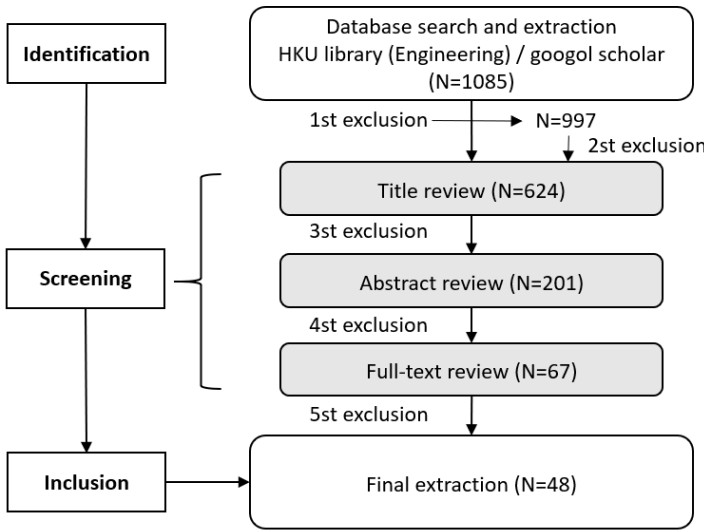

**Figure 2: Preferred Reporting Items for Systematic Reviews and Meta-Analyses (PRISMA) framework showing the literature screening process.**

## 3 Methods for urban surface water flooding

As introduced above, urban flood models are classified by four types. Selected studies and relevant models are listed in Table 3. The core of these models mainly lies in how to quantify the three key components of flooding in urban systems – rainfall, surface runoff and drainage flows – as shown in Figure 3. The key methods used in each type of model are summarised below.

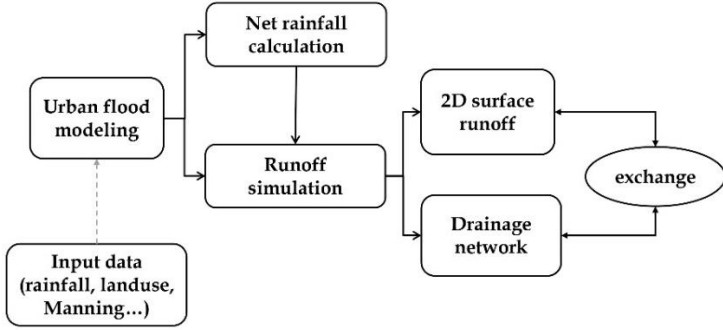

**Figure 3: Key components of urban flood modelling.**

**Table 3: Summary of the characteristics pertinent to urban flood models.**

| Representative models | Model Equations | Acceleration Method | Rainfall-runoff | drainage network module |
|---|---|---|---|---|

| | | | | | |
|---|---|---|---|---|---|
| Drainage network models | Djordjevic et al. (1999) | 1D Saint-Venant equations | No | No | Yes |
| | Schmitt et al. (2004) | 1D Saint-Venant equations | No | No | Yes |
| SWE-based models | FloodMap (Yu, 2010) | 2D diffusive wave model | MPI | Yes | No |
| | LISFLOOD-FP (Bates et al. 2010) | Inertial formulation of 2D SWEs | OpenMP | No | No |
| | UIM (Chen et al. 2012) | 2D diffusive wave model | No | Yes | HEC-1 |
| | P-DWave (Leandro et al. 2016) | 2D diffusive wave model | OpenMP | Yes | 1D Saint-Venant equations |
| | PRIMo (Sanders et al. 2019) | 2D SWEs based on upscaled grids | single process multiple data parallel algorithm | Yes | No |
| | RMA (Rao 2005) | simplified 2D SWEs | MPI | Yes | No |
| | UPFLOOD (Huang et al. 2019) | 2D diffusive wave model | No | Yes | No |
| | TRENT (Villanueva et al. 2006) | 2D full SWEs | No | No | No |
| | Porosity-based models (e.g. Guinot et al., 2017; Bruwier et al., 2017) | 2D full SWEs with porosity in coarse grids | No | No | No |
| | CityCAT (Glenis et al. 2018) | 2D full SWEs | Amazon Cloud | Yes | MFP-model (Bourdarias et al. 2011) |
| | Hou et al. (2018) | 2D full SWEs | GPU | Yes | No |
| | HiPIMS (Xia et al. 2019) | 2D full SWEs | GPU | Yes | 1D Saint-Venant equations |
| | Henonin et al (2015) | 2D full SWEs | No | Yes | No |
| | Liu et al. (2019) | 2D full SWEs | GPU | Yes | No |
| | Rajib et al. (2020) | Inertial formulation of 2D SWEs | OpenMP | No | No |
| Hydrogeomorphic approaches | Nardi et al (2018) | hydrogeomorphic paradigm | No | Yes | No |
| | GeoFlood (Zheng et al. 2018) | hydrogeomorphic paradigm | No | Yes | No |
| Other methods | CADDIES-caflood (Guidolin et al., 2016) | cellular automata | GPU | No | No |
| | Bermudez et al. (2018) | ANNs | No | Yes | Yes |

## 3.1 Drainage network models

The drainage network is the key infrastructure that drains stormwater runoff in urban areas. Inundation in urban surfaces is always caused by the surcharge, which means the capacity of the drainage network cannot support the urban surface runoff. Therefore, drainage network models are often used for simulating urban stormwater runoff when detailed pipe-network data is available (Lee et al., 2019). The main goals of such models are to simulate the streamflow in the underground drainage network and to provide the flow hydrograph at the outlets of urban catchments or sub-catchments.

In a confined channel or in a pipe, the flow is generally considered to be one dimensional. Drainage network models solve equations derived by ensuring mass and momentum conservation between two cross sections $x$ apart, which yields the well-known one-dimensional Saint-Venant equations. In the urban drainage network, as the pipeline discharge increases or decreases, the pipeline flow may change from open channel flow to pressurised flow back and forth. Therein, the open channel flow has a free surface, and the pressure can be approximated as satisfying the assumption of hydrostatic pressure. But the open channel flow control equation is used for the simulation of open channel drainage flow (Eq. (1) and (2)). However, the pressurised flow does not have a free surface, and the pressure no longer meets the hydrostatic pressure assumption. Thus, a set of modified equations (Eq.3 and 4) are used. These models were also called '0-term' models (Neelz et al., 2013) corresponding to the full-term shallow water-based models.

The mass and momentum conservation equations for open channel flows with water level $Z$ and flow discharge $Q$ as variables are as follows:

Conservation of mass $\quad \frac{\partial Z}{\partial t} + \frac{1}{B}\frac{\partial Q}{\partial x} = 0$ (1)

Conservation of momentum $\quad \frac{\partial Q}{\partial t} + \frac{\partial}{\partial x}\left(\frac{Q^2}{A}\right) + gA\frac{\partial Z}{\partial x} + gAS_f = 0$ (2)

where $Q$ represents the flow discharge, $Z$ represents the water level, $S_f$ represents the friction slope, $A$ is the flow cross-section area, $B$ is the width of water surface, $t$ is time, and $g$ is the gravitational acceleration.

The mass and momentum conservation equations for pressurised flows with piezometric head $H$ and flow discharge $Q$ as variables are as follows:

$\frac{\partial H}{\partial t} + \frac{a^2}{gA}\frac{\partial Q}{\partial x} = 0$ (3)

$\frac{\partial Q}{\partial t} + \frac{\partial}{\partial x}\left(\frac{Q^2}{A}\right) + gA\frac{\partial Z}{\partial x} + gAS_f = 0$ (4)

where $H$ is the piezometric head, and $a$ is the wave velocity.

A drainage network model is generally coupled with an urban hydrological or hydraulic model to quantify surface runoff on the urban surface. For example, urban hydrological models, such as the Urban Drainage and Sewer Model (MOUSE) (DHI,

2002), EPA SWMM (Rossman, 2010) and other research models (Schmitt et al., 2004; Simoes et al., 2010), are frequently implemented or redeveloped to simulate urban rainfall-runoff and flood overcharge from drainage maintenance holes. However, these models cannot reproduce the dynamic processes of urban inundation. To simulate urban surface water flooding, dual drainage models that combine drainage networks with urban street networks have been developed (Djordjevic et al., 2005; Simoes et al., 2010). During flood events, drainage pipe flows generally produce surcharge at urban drainage

manholes, resulting in flooding on the street surface. Dual drainage models take into account both flows within the drainage system and surcharge over the streets during intense rainfall. This type of model has two interactive modules: (1) an underground module that consists of a sewer system with known manholes, inlets, and control structures and (2) a surface module including flow paths, retention basins in local depressions or other artificial control structures (brinks, ponds) made of channels (Mark et al., 2004; Djordjevic et al.,2005). In essence, the 1D model represents the surface flow path (mainly streets)

on top of a 1D-pipe-flow model, with exchanges through gully port, catch basin or other coupling junctures. Then, urban flood risks are assessed based on a dual-drainage-modelled output hydrograph, the water remaining in the drainage network, and water depths on street surfaces with the aid of the analytical capabilities of Geographic Information System (GIS). However, 1D model cannot be used to reproduce the rainfall-flood process as it cannot be one-way flow in the whole urban. It is considered over-assumed when treating surface flow as 1D channelled flow, which clearly limits its application.

**3.2 Shallow water-based models**

In recent decades, high-resolution Digital Elevation Model (DEM) and Digital Surface Model (DSM) data with more detailed spatial information are increasingly available. Hydrodynamic models built upon SWEs have demonstrated strong capabilities in providing more detailed flood information in urban areas, such as distributed floodwater depths and velocities. SWE-based models have been frequently applied to fluvial and coastal flooding but were recently refined for urban surface water flooding

(Gomez et al., 2011; Xia et al., 2017).

Urban surface floods generally do not have a fixed path and direction and are generally widespread in two-dimensional or even three-dimensional in case of large water depth. The motion of incompressible viscous fluid can be described by the Navier-Stokes equation. Due to the heavy calculation burden and complicated algorithm design, it is obvious that traditional hydrological methods and one-dimensional hydrodynamic models cannot effectively deal with such problems. In practical

applications, the equations are generally simplified appropriately according to the specific characteristics of the flow conditions. The urban surface runoff is generally shallow water flow, that is, the movement of shallow water with free surface under the action of gravity. The two-dimensional shallow water equations can realize the accurate calculation of urban flood inundation, which can be obtained by simplifying the Navier-Stokes equations in the vertical direction. It can meet most of the application requirements in engineering practices and has been widely used in shallow water research. Based on comprehensive

reviews of SWEs-based models for both fluvial flows and urban surface water, we summarise the governing equations as below (Audusse et al., 2004; Liang and Marche, 2009; Toro, 2013).

Conservation of mass $\frac{\partial h}{\partial t} + \frac{\partial q_x}{\partial x} + \frac{\partial q_y}{\partial y} = R + E$ (5)

Conservation of momentum $\frac{\partial q_x}{\partial t} + \frac{\partial}{\partial x}(uq_x) + \frac{\partial}{\partial y}(vq_x) = S_{bx} + S_{fx}$ (6)

$$\frac{\partial q_y}{\partial t} + \frac{\partial}{\partial x}(uq_y) + \frac{\partial}{\partial y}(vq_y) = S_{by} + S_{fy}$$ (7)

165        ①     ②     ②     ③   ④

where $x$ and $y$ are the two Cartesian directions, $t$ is time, $h$ is the water depth, $q_x$ and $q_y$ are the x and y components of the discharge per unit width, $u$ and $v$ are the $x$ and $y$ components of the flow velocity, $z$ is the bed elevation, $g$ is the gravitational acceleration, $R$ is the source or sink term representing net rainfall intensity (runoff term), rainfall intensity is accumulated into the mass equation as a mass input, and $E$ is a pipe-surface exchange flow term to connect the flow between 2D surface runoff

and 1D drainage network flow. $S_b$ and $S_f$ are the bed slope source term vectors and friction effect source term vectors, respectively. The numbers below the equation represent the different terms of the shallow water equations: ① local acceleration, ②convective acceleration, ③ pressure + bed gradients and ④ friction.

### 3.2.1 Simplified 2D shallow water models

SWE-based models have been explored in recent decades for improving both the efficiency and accuracy of simulations.
However, there are challenges to modelling urban surface water flood using this approach, due to the complex and irregular terrains and the lack of sufficient input data, especially for large-scale modelling (Leandro et al., 2016). Since full shallow water models are computationally expensive, some studies suggest omitting or approximating less significant terms in Eq. (5–7) to reduce model complexity and save computational costs (e.g. Yu and Lanes, 2006; Bates et al., 2010; Martins et al., 2017a; Sanders et al., 2019). The simplified SWE-based models include 2D diffusion wave models that neglect the inertial (local
acceleration term ①) and advection (convective acceleration term ②) terms, '3-term' models (Bradbrook et al., 2004; Bates et al., 2010), and 2D kinematic wave models that omit pressure terms (③) as well, also called '2-term' models (Hunter et al., 2007).

The kinematic wave model was initially developed for fluvial flooding with deeper water (e.g. Singh, 2001; Hunter et al., 2007). However, major assumptions have to be made when being applied in urban surface water flooding, which has relatively
shallower water. Such 2D kinematic wave models can give a reasonable level of accuracy only for simple flow regimes (Zhang, 2014). So, the diffusion wave approximation, introduced by Cunge et al. (1980), was considered a more practical simplification. The diffusion wave model is also called the zero inertial model. Yet, the lack of inertial terms may raise issues of model accuracy and stability, e.g. the control of calculation time. When the constant time step is not small enough, 'chequerboard'-type oscillations will be generated, where water in one particular cell drains into the adjacent cell in a single

large time step and flows back in the next time step (Hunter et al., 2005; Zhang, 2014). This results in a loss of water mass, which affects the accuracy in predicting 'shallower' urban surface flooding driven by rainfall (Huang et al., 2019).

To avoid instability, a flow limiter is generally used in a diffusion wave model to prevent water leakage from a given cell in a single time step. Hunter et al. (2005) proposed an adaptive time step as an alternative to the flow limiter. A similar method is also applied in P-DWave (Leandro et al., 2016). Later work by Hunter et al. (2008) found that a partial inertial model (also

known as a local inertial model or simple inertial model) can be set up by including the local acceleration term, which allows the use of a larger time step and eliminates severe oscillation in the water. This assumes that flow advection is inconsequential in floodplains, so the convective acceleration term can be excluded. Several other studies (e.g. Fewtrell et al., 2011; Almeida and Bates, 2013) validated the simplified approach through numerical cases against field data and analytical solutions. Almeida and Bates (2013) applied the partial inertial approximation model to several flow problems, showing that the scheme could

provide relatively accurate and efficient results. Several other models have been developed based on a similar theory (e.g. Yu, 2010; Leandro et al., 2016).

Applications of these models to real-world events indicate that inundation in urban areas could be adequately modelled if considering inherent uncertainties of various data input and model conceptualisation (Willis et al., 2019). Some studies (e.g. Leandro et al., 2006; Henonin et al., 2015) developed sub-grid models that improve calculation efficiency while ensuring

certain accuracy by implementing relatively coarse numerical solutions. To increase model accuracy, Huang et al. (2019) proposed a calculation of the tangential gradient at the cell edge to improve the accuracy. Hunter et al. (2005) also mentioned that owing to the stricter time step control needed for stability, the effect of diffusion wave approximation in saving computational time is not evident in high-resolution simulations.

### 3.2.2 Full 2D shallow water models

Simplified SWE-based models are less computationally expensive compared to full 2D shallow water models because of the simplification or omission of certain hydraulic processes. However, when facing complex flow regimes, e.g. transcritical flows, supercritical flows or shock-like flow discontinuities, full 2D shallow water equations, Eq. (5–7), are essential to reproduce the full flood dynamics. Full SWE-based models have presented a potential for appropriate approximations in specific situations and are commonly used for surface water issues (Sanders et al., 2008; Liang and Marche al., 2009; Ferrari et al.,

2019). 2D dynamic wave models have been considered the only option to predict the backwater effects accurately and hydraulic-hydrological discontinuities (Gomez et al., 2011). Fluvial flood models based on full SWEs were developed by solving Eq. (5–7) in past decades. Schemes based on the finite-volume method and approximate Riemann solvers have gained recent attention with their adequate handling of discontinuities in the flow field. The good track record of full-SWE research provides a solid basis for extending its application to urban surface flood modelling. As indicated in Figure 3, urban surface

water flooding also includes rainfall, infiltration and flow exchange flow terms. Thus, further efforts are needed to achieve urban surface flood modelling by using full SWEs.

There are numerical challenges when applying fluvial flood dynamic model to urban surface water flooding:

- representation of urban terrain

- appropriate processing of shallow surface water in irregular bed

• balance of computational efficiency and accuracy

In recent years, studies have been undertaken to overcome these numerical challenges. For example, Xia et al. (2017) introduced a numerical scheme for modelling overland flows over complex bed terrains by developing methods to maintain numerical stability and accuracy. Different from natural catchments, urban areas contain complex topographic features and underground infrastructures that heavily affect urban inundation. For this, some studies proposed a porosity-based SWE model to generalise the effects of dense urban buildings (Sanders et al., 2008; Kim et al., 2015; Guinot et al., 2017; Bruwier et al., 2017; Ferrari & Viero, 2020). Bruwier et al. (2017) proposed an approach to determinate the porosity parameters reflecting the different characteristic size of obstacles. This method can obtain speed-up values between 10 and 100 while the errors on water depths remain low. Ferrari & Viero (2020) presents an algorithm to automatically extract the spatial distribution of the porosity parameters by geometrical information in complex urban areas. The schemes were shown to provide reasonably good results in experimental and city case studies (Ferrari & Viero, 2020, Viero, 2019; Mel et al., 2020) The studies on porosity-based models mostly focus on numerical schemes for solving urban flood dynamics driven by upstream inflows, rather than direct rainfall. Few large-scale applications in surface water flooding driven by rainfall have been studied. Moreover, sophisticated urban flood models have been developed recently by including detailed urban surface buildings and underground drainage systems. For example, Glenis et al. (2018) presented a full 2D SWE-based model that includes not only a module simulating pipe flow but also representative buildings and urban infrastructures based on high-resolution DEM.

For large-scale applications of full 2D SWEs-based models, high-resolution data, such as DEM/DSM, is required to represent urban building blocks and surface conditions, since street-level or meter-scale modelling implies a large number of computational grids. Parallel algorithms and speed-up techniques to overcome this dramatic increase in computational costs have been the focus in the last decade. Numerous studies have justified that a GPU-based parallel algorithm is capable of speeding up a flood model by over ten times (e.g. Kalyanapu et al., 2011; Vacondio et al., 2014; Smith and Liang, 2013). Particularly for catchment-scale flooding, Xia et al. (2019) developed a GPU algorithm to accelerate a flood model and successfully reproduce the rainfall-inundation process in Eden Catchment caused by the 2015 storm Desmond (2500 km2 with resolution of 5m). Hou et al. (2018) presented a GPU-based urban flood model to reproduce a flood event under 100 years design storm event in Morpeth town, UK. Many other studies also indicate real-time modelling of flooding is possible in a domain with over 100 million grids by using GPU accelerated algorithms. Moreover, cloud computing has been used to accelerate the sophisticated urban flood model, e.g. Glenis et al. (2018). In summary, to achieve large-scale urban modelling, significant assumptions are needed with a simplified model of a coarse scale, as stated in Section 3.2.1, while full SWEs-based models are frequently based on GPU or cloud computing.

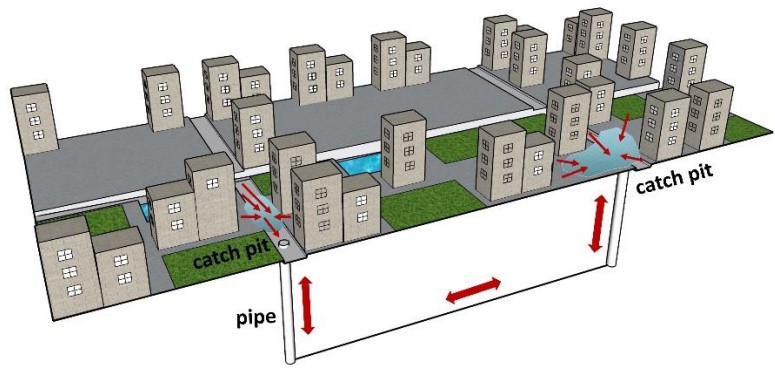

**Figure 4: 1D drainage network coupled to 2D surface flow model.3.2.3 Drainage network coupled to the urban surface model**

In SWE-based dynamic models, the effects of drainage networks are mostly either neglected or over-assumed by a constant drainage capacity. Even though the studies mentioned above have developed advanced urban flood models, drainage flows were not taken into consideration (e.g. Sanders et al., 2008; Xia et al., 2019). However, the drainage network plays a crucial role in draining urban runoff, which will re-distribute surface water inundation during flooding (demonstrated in Figure 4). The omission or over-assumptions may result in the mis-simulation of surface water depths, inundation coverage and duration, particularly at a localised scale. In fact, the drainage network should be especially taken into account when the duration of the inundation represents a key variable for a correct flood risk evaluation. Although still in its infancy, coupled modelling of drainage flow and urban surface water is increasingly attracting attention using 1D pipe flow and 2D surface flow (Seyoum et al., 2011; Chen et al., 2015; Bazin et al., 2014; Leandro and Martins 2016; Martins et al., 2017; Li et al., 2020). Such models represent drainage flow by a 1D pipe model and quantify the effluents and influents via manholes using the weir or the orifice equations (Bazin et al., 2014). Urban surface water is reproduced by these coupled models with a 2D shallow water model, allowing the simulation of the time series of flow spreading across the urban area.

Based on the existing studies, there are three commonly used methods to characterise drainage flow in urban flooding simulation:

(1) Using a runoff term R in one of three typical techniques: ① reducing the rainfall rate i in the runoff term R in all of the study area, which is i minus Ip, a constant value representing the drainage capacity of the pipe network; ② an additional value is added to the natural infiltration I of the soil in the runoff term R of the entire study area to represent the capacity of the drainage network, which is I plus Ip; ③ the drainage capacity of the pipe network is accumulated to the actual layout range (e.g. Environment Agency, 2013; Hou et al., 2018)

(2) Exchanging discharge through coupling links by various approaches including: orifice and weir equations (Chen et al., 2007; Huang et al., 2019; Rubinato et al., 2017; Rubinato et al., 2018;), inlets or gullies capacity (Leandro et al., 2009) and the displacement of manholes covers (Chen et al., 2016).

(3) Applying 1D Saint-Venant equations, which is the most widely used and accurate of the three techniques. The surface flow spreading process can be calculated by 2D model part obtaining results such as water levels and velocities. Then bidirectional components exchanging is allowed in the junction with the coupling 1D drainage model to realize the reproduce of the urban storm-flood process (Carr & Smith, 2006; Mark & Djordjevic, 2006; Li et al., 2020). A model concept is demonstrated in Figure 5. The horizontal coupling means components exchanging between the 2D surface cell and the drainage cell in horizontal direction, and vertical coupling means the discharge redistribution in the drainage cell, to calculate surcharge from the urban surface into the drainage or backflow from the pipe.

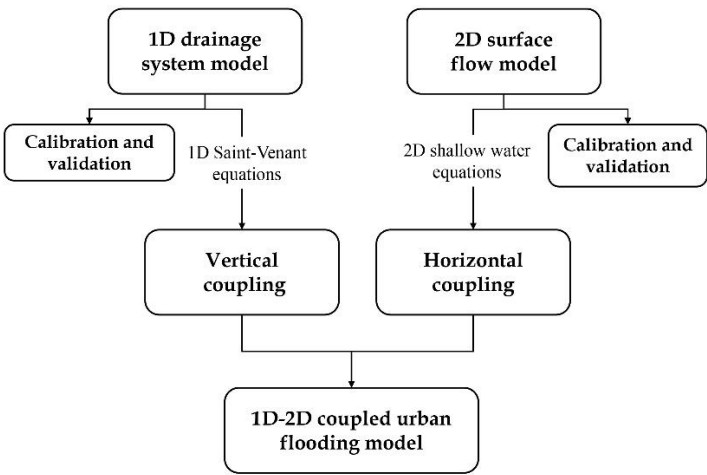

**Figure 5: Coupled 1D-2D urban flood model.**

The first method does not consider the influence of the surface morphology of the pipe network system and the process of surface runoff flowing into the pipe network. Therefore, the rainwater is reduced before reaching the road surface, which is inconsistent with the actual process. Also, surface runoff into the stormwater inlet is not constant, and the simplification of the first method can be problematic. For a more accurate simulation of urban flooding, the second method has been widely applied in the last decade.

Recently, the third coupling approach has been increasingly investigated. For example, Martins et al. (2017b) validate a 2D shock-capturing flood model coupled with a 1D unsteady pipe flow solver based on both quadrilateral-structured and triangular-unstructured mesh-types. Li et al. (2020) proposed a junction simulation approach, instead of the traditional method solving the continuity equation, which is coupled with the widely used two-component pressure approach (TPA) and led to a new integrated drainage network model. The new 1D-2D coupled drainage network model was validated against an

experimental and several idealised test cases to demonstrate its potential for efficient and stable simulation of flow dynamics in drainage networks. However, there are physical complexities and uncertainties for applications in real-world events.

### 3.2.4 Hydrological coupled to hydrodynamic urban flood model

Even with the rapid development of SWE-base models, it is still challenging for both simplified and full models to be applied in large-scale events. A coupled modelling framework has been developed for large-scale flood modelling (Liu et al., 2019; Rajib et al., 2020). More specifically, as shown in Figure 6, the model framework divides the hydraulic structures, river reach with complicated flow conditions and urban inundated areas into hydraulic zones (gridded while zone), while the rural domain is defined as hydrological zones (blue zone). In hydrological zones, selected hydrological methods are applied for flood routing, and the SWEs are used to simulate the surface water dynamic process in hydraulic areas. The green area in the Figure 6 is the boundary area, where bidirectional components exchanging between hydraulic and hydrological areas is allowed to update the calculation.

There are two common hydrological and hydrodynamic coupling approaches, external coupling and internal coupling. For external approaches, the results of hydrological models, such as hydrographs, can be applied as upstream or lateral boundary conditions for hydraulic models. The one-way and two-way transition are all allowed in this method. It is suitable and has a wide range of application urban catchment with complex river network system (e.g. Lian et al., 2007; Mejia & Reed, 2011; Kim et al., 2012; Liu et al., 2019). For internal approaches, the hydrological models and hydraulic models are calculated separately and update the information at the shared boundary with a certain calculation time steps interval. The main intention of such model frameworks is to predict urban flooding from the perspective of broader catchment scales efficiently with reasonable accuracy. To a certain extent, such model frameworks also reduce the uncertainty effects of hydrodynamic models when applying in upstream rural catchments.

### 3.3 Hydrogeomorphic approaches

Apart from the models reviewed in section 3.1 and 3.2, an emerging method, the hydrogeomorphic approach, has been recently developed for flood hazard management and mapping (Nardi et al. 2013; Zheng et al. 2018). Different from the physical-based models above, hydrogeomorphic approaches are based on the concept of fractal river basins or hydrogeomorphic theories, and floodplains are identified as unique morphologic landscape features. The inundation area is clearly recognisable by using terrain data analyses (Di Baldassarre et al., 2020). It is not necessary to estimate the synthetic flood hydrograph and the floodplain can be determined consistently in different regions. Specifically, a simplified hydrologic analysis can provide the elevation thresholds of the potential inundated grid under different discharge condition then identified the floodplain cells for a different return period (Nardi et al., 2018).

More recently, the increasing availability of high-accuracy Digital Terrain Models (DTMs) provide new opportunities for morphometric analysis of floodplain mapping. And this floodplain delineation method has been of significant interest because

of the low requirements of time series data and high computational efficiency. It is being developed to apply in global and
large-scale catchment fluvial flood mapping and has also been applied in urban areas in recent years (Brown de Colstoun et
al., 2017; Nardi et al., 2018). However, its model concept implies that such a method can predict an approximate inundation
area but cannot simulate the dynamic processes of flooding (time series of water depths, velocities, etc.) that is vital for risk
assessment.

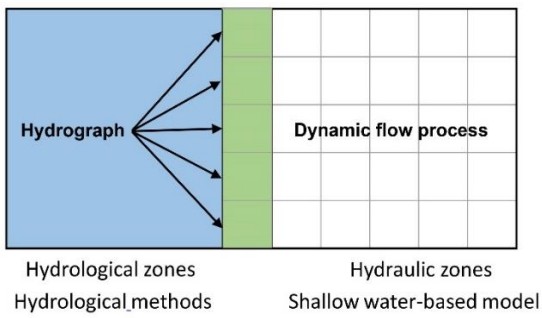

**Figure 6: Divided study area of hydrological and hydrodynamic coupled urban surface model.**

### 3.4 Other methods

*Cellular automata models*. Recently, several studies have evaluated an alternative urban flood modelling approach instead of
solving the SWEs, and the cellular automata (CA) approach shows the great prospect in urban inundation simulation (Dottori
& Todini, 2011; Ghimire et al., 2013; Guidolin et al., 2016). The CA models represent the discrete simulation area with grids.
Apart from the regular properties, such as the state of each cell, distribution of surrounding cells and discrete time step, there
are a set of transition rules. Based on the previous state of the cell and neighbouring cells, the rules control the evolution of
each cell state (Teng et al., 2017). Austin et al. (2014), successfully developed a series of CA models (e.g. CA1D and BCA1D.)
to simulate sewer network flow with various transition rules. Compared with the traditional 1D hydraulic models, these
simplified CA models can produce reliable results with high computational efficiency. Guidolin et al. (2016) developed a
weighted 2D CA inundation model, CADDIES 2D flood model WCA2D, that obtained the results consistent with the accuracy
of those diffuse wave model. CA algorithm is suitable for parallel computing as the hydraulic properties evolution of each cell
only requires the state of it and the surrounding cells at the previous calculation time step. However, CA approaches are newly-
developing in the field of urban flood simulation just emerging in urban applications. It is shown that CA models can be
valuable tools for flood simulation. However, there is only limited literature and most of the tests are ideal cases. Besides, it
showed less accuracy in reproducing two-dimensional flow dynamics in respect to a model based on the full shallow water
equations. So, in this concern, it still needs to be further verified in 1D and 2D practical applications (Guidolin et al., 2016).

*Artificial neural network models*. As high-quality data becomes more available, data-driven models, such as artificial neural
networks (ANNs), have emerged and can be useful in view of their merits in computational costs and speeds. An ANN is an
approximation technique that has been widely used in the water resources field (Yaseen et al., 2015; Wolfs and Willems, 2014;

Lin et al., 2020). The applications to urban surface water flooding are still fairly scarce. However, for example, Bermudez et al. (2018) established a model based on two different ANNs to simulate the urban sewer-flood procession, and Berkhahn et al. (2019) presented an ANN-based model to predict the maximum water levels during a flash flood event in an urban river or urban area. The computation time of these models are significantly less than their predecessors, and the results are supported with abundant field data, confirming the importance of having high-quality data.

## 4 Advantages and limitations

Each model has advantages and limitations when applied in urban flooding, which will be discussed in this section. A comparative summary is listed in Table 4.

**Table 4: Comparative summary of the relative advantages and disadvantages of different models.**

| Method | | Features | Strength | Limitation | Suitability |
|---|---|---|---|---|---|
| Drainage network models | | • based on the 1D Saint-Venant equations<br>• focus on the underground drainage network streamflow simulation<br>• simplify the 2D surface flow calculation | • computationally efficient<br>• suitable for various temporal and spatial scale evaluations<br>• quantification of flow in drainage systems | • coarse spatiotemporal resolution<br>• no/little flow dynamics<br>• verification is data and time demanding (large number of drainage nodes and pipes) | • drainage system design and evaluation<br>• quantification of outflow of urban catchments<br>• a reference for other inundation models<br>• a tool coupled with surface water model |
| 2D SWE based models | Simplified SWEs-based models | • based on the 2D simplified shallow water equations (diffusion wave or kinematic wave)<br>• focus on the 2D surface flow simulation<br>• ignore or simplify the process of underground drainage flow | • dynamic simulation of urban flooding within relative cheaper computational cost | • cannot capture shock flood wave<br>• Less numerical accuracy compared with full SWEs<br>• no pipe flow consideration | • urban flood mapping without high requirement on detailed flow dynamics<br>• faster urban flood simulations |
| | Full SWEs-based models | • based on the full 2D shallow water equations<br>• focus on the 2D surface flow simulation<br>• ignore or simplify the process of underground drainage flow | • full dynamic simulation<br>• shock-captured<br>• be able to simulate the flow-infrastructure interactions | • computationally expensive<br>• high requirement for data inputs<br>• no pipe flow consideration or over assumption | • quantification of local urban flood dynamics without pipe<br>• urban flood model in urban areas with high- |

| | | | | quality DEM/DSM • design and evaluation of flood infrastructures |
|---|---|---|---|---|
| Coupled with drainage network | • coupled modelling of drainage flow and urban surface water (SWEs) • underground drainage flow calculated in different methods to re-distribute surface water inundation | • can simulate drainage floods • have potential to simulate urban flooding more accurately | • computationally expensive • requirement of good-quality data input | • quantification of drainage flooding • simulation of local urban flood dynamics with pipe • urban drainage design and evaluation |
| Coupled with hydrological methods | • divide into hydraulic zones (SWEs to simulate the surface water dynamic process) and hydrological zones (hydrological methods for flood routing) • components exchanging between hydraulic and hydrological areas is allowed | • computational efficient • plus the strength of its coupled hydraulic models | • currently no pipe consideration • plus the limitation of its coupled hydraulic models | • large-scale catchments where natural areas have equal runoff contribution with urban areas |
| Hydrogeomorphic approaches | • based on the concept of fractal river basins or hydrogeomorphic theories • use terrain data analyses to recognize inundation area | • Less sensitive to data scarcity and time series data unnecessary • Computationally efficient | • cannot include the role of infrastructures and altered geomorphic signature in urban areas • uncertainty of empirical data • no flow dynamics representation | • preliminary identify inundation areas • a reference for physically representative models. |

## 4.1 Drainage network models

Drainage network models, particularly coupled with hydrological methods, are highly computational efficient owing to their simple structures; thus, they show merits in applications that focus on flood-related hydrological analysis in an urban catchment and have low requirements for the representation of full hydrodynamics. The fast operation of this model type makes it suitable for large-scale simulations with various temporal scale evaluations that require thousands of longer-run simulations with little detail of the flow behaviours. Inclusion of pipe network and manholes in models can help to estimate potential surcharge,

which can be used as a reference for other inundation models, e.g. 1D street network models or 2D surface water models. The

surface runoff simulation of these models is mainly based on the conceptual hydrological models, or grey-box models. Although urban rainfall-runoff processes can be properly estimated by these models through linking a drainage network, the over-assumption of flow dynamics does not allow for the simulation of surface flow inundation, which is a key indicator for risk assessment. Thus, there will be a lack of detailed spatial dynamic information of urban surface waters, such as flood depth and velocity. Moreover, this type of model usually solves the 1D equations for drainage flows. In the city scale, the drainage network system is often complex with a large number of drainage nodes and pipes. Although automatic GIS procedures can improve efficiency to a certain extent, it is still time-consuming data-demanding to set-up and verifies this type of model. For operational flood management, there is currently a high demand for thorough, detailed spatial information, such as the depth and velocity of surface flow in each street and accurate data for each residence. Drainage network models cannot meet above requirements for urban surface water flooding.

## 4.2 Shallow water-based models

A power of work has been devoted to developing 2D shallow-water-based numerical models in recent decades. These include simplified and full SWEs models, models coupled with drainage network and even hydrological and hydrodynamic coupled models. In the above overview, SWE-based models have proven to be capable of reproducing surface flow reasonably well for flooding in urban areas, accurately predicting velocity, flood extent and water level.

High-resolution topographical data, such as LiDAR data, DEM or DSM, are now becoming available and can have a fine resolution of about 20 cm or even less. Several studies have developed applications of SWE-based models in a variety of scales by developing new algorithms or new model framework and utilisation of high-resolution topography data sets. These models are advancing current practices to make practical simulations of urban surface water inundation driven by extreme rainfall. Experimental and field in-situ data has been always considered as essential supporting source for flood model validation. In this concern, some recent studies have specifically developed physical models and approaches to gather detailed experimental/field data to support flood model verification (Rubinato et al.,2013; Lopes et al., 2015; Gómez et al., 2019; Martins et al., 2018). For example, Rubinato et al. (2013) and Lopes et al. (2015) collect hydraulic data form a physical model and apply the data to verify the performance of a numerical model. Although there are some, good-quality benchmark dataset is still a lack in research community for urban flood model validation.

Simplified and full SWE-based models. Compared to full SWE-based models, simplified SWE-based models have been typically used in larger-scale urban flood modelling with coarser grids and simplified treatment of urban features (e.g. city-scale, even continental scale) because of the relatively low computational costs and the capability in simulating surface water dynamics (e.g. Yu et al., 2016; Wing et al., 2017). However, the inherent model conceptualisation implies that these models cannot capture the shock wave adequately. They may be not an ideal choice when simulating detailed urban flood dynamics with infrastructures.

Recently, sophisticated full SWE-based models have also been applied to city-scale flood modelling with the advances of accelerated algorithms (e.g. Glenis et al., 2018; Xia et al., 2019; Sanders and Schubert, 2019). Such models are capable of simulating the full surface water dynamics. However, they require high-quality data, which could result in high uncertainty depending on the data source (Willis et al., 2019). Both simplified and full SWE-based models have weaknesses, and further improvements are current challenges:

- The concentration of buildings in urban areas plays a vital role in the magnitude of inundation and its hydrodynamics; however, current SWE-based models either ignore its effects or make over-assumptions. Some attempts with the inclusion of urban features mainly focus on small-scale validations of numerical schemes, whereas larger-scale applications have been hardly studied.

- The computational cost of 2D full SWEs-based models is especially significant for high-resolution modelling, which is necessary for representing urban features, such as buildings. While sometimes acceptable for events-based applications, this price is impractical for real-time simulations that serve for early flood warnings. Despite the rapid development of accelerated computing techniques, 2D shallow water models are still considered not feasible for calculations in large-scale catchments with fine resolution grids, because the simulation time required may be prohibitive (Neelz and Pender, 2013). It is a current challenge to balance computational efficiency with accuracy by either optimising numerical algorithms or establishing reasonable model structures.

***Drainage network coupled to the urban surface model***. As a key feature of urban catchment, the drainage network has also been ignored or generalised with over-assumptions as discussed above. Although there are alternative approaches to treat pipe flows, and coupled hydrodynamic models are emerging, they are still in its infancy for real-world applications (Mignot et al., 2019). Theoretical or experimental-scale testing for coupled hydrodynamic models may be successful (Li et al., 2020), but improved model performance has not actually been reported when including a drainage flow model into a real-world surface water flood modelling. Also, the drainage network data and unclear coupling mechanisms of pipe flow and surface water heavily limit its large-scale applications in real-world applications. Nonetheless, the detailed model would be applicable to the detailed flow dynamic investigation in pipe and urban surface systems at a localised scale.

***Coupled hydrological and hydrodynamic urban flood models***. By dividing the simulated domain into hydrological and hydraulic zones, these models alleviate the issue of expensive computational costs that pure hydrodynamic models encounter in large-scale applications. Therefore, they are suitable for predicting urban flooding in large-scale catchments, where natural areas have similar runoff contributions to urban areas. Similarly, with pure hydrodynamic models, existing studies also either neglected or made significant assumptions on the treatment of pipe flows.

## 4.3 Hydrogeomorphic approaches

Hydrogeomorphic approaches can identify the inundation area directly from the topography. Thus, it requires much less computation time and no time-series data. However, there are still many limitations for the application of these.

- Only approximated inundation area can be obtained. Flow dynamic information (water depth, velocity, et al.) cannot be provided.
- The effects of recent anthropogenic modifications on a floodplain or urban surface features will increase uncertainty in a floodplain map. There is no effective way to characterise infrastructures and the altered geomorphic signature in highly urbanised areas.
- The resolution and accuracy of DTMs, as well as terrain data processing and analysis, are challenging.
- Empirical data related to flood stage are needed to determine the floodplain flow depth scaling relationship. These data have great uncertainties, such as availability and quality, which directly affect whether to get the accurate floodplain area.

However, flood mapping can be considered as a complement to physically-based hydrodynamic modelling. For example, using an ungauged condition to preliminarily identify inundation areas can provide references for physically representative models.

## 5 Future challenges

### 5.1 Refinement of SWEs-based models

In view of the current advances of urban flood models, there are still deficiencies in improving model reliability and efficiency. Urban areas have many complex underlying surface characteristics, and in reality, when the capacity of drainage networks is insufficient, the pipe flow will over-charge to the ground surface. At present, simulation methods of the exchange of pipe flows and surface waters are only focused on local-scale modelling. Some existing numerical models often directly use empirical formulas or simplified methods that are still lacking in stability and accuracy. Therefore, mechanisms between pipe flow and surface water and their modelling approaches would help simulate drainage flooding in urban areas. This could be accomplished by integrating drainage network models with overland flow routing models as some studies have done, but further refinement is needed. Moreover, the question of which model conceptualisation is more appropriate for urban surface water flooding is still unanswered and in need of further investigation with the support of high-quality data.

The spatial heterogeneity of urban catchments is typically more profound, and surface water depths are generally shallower. Both pose numerical challenges in solving SWEs over frictional and extreme, irregular terrain. Although the application of parallel computing technology to improve the computational efficiency of the model has become a trend, efficient urban flood simulation and even real-time flood prediction with better resolutions are still difficult to achieve. In the light of the high

computational costs for large-scale modelling at high-resolution, more accurate and faster model algorithms are urgently needed. This is critical for achieving a city-scale urban flood prediction in real-time.

## 5.2 Data-driven approaches

With the continuous improvement of remote sensing technology, the data become more readily available. A data-rich environment also encourages model calibration, validation and assimilation. In practical terms, the accuracy of terrain data

obtained from modern LiDAR system has met the requirements for surface flow simulation, but it is necessary to fuse such dataset with digital map data of buildings and land use to realize the maximum development and utilization of contained information. Furthermore, these terrain data are readily available, and topographical data with a grid scale of roughly 30 m at best hardly meets accuracy requirements for urban flood models.

Model calibration is an essential way to reduce uncertainty over model parameters, but to this day, such data has been scarce

for urban areas. Despite the frequency of urban floods, field observations during urban flooding are rarely available for model calibration and validation. Calibration data will be the key factor constraining the future development of urban flood inundation models. Some effective methods or tools are therefore urgently needed to infer from these limited data sources, extending the quantity and range of typically available calibration-validation data. Development of physical urban flood models is an option to gather benchmark data for urban flood modelling as some researchers have done (e.g., Rubinato et al., 2017). Nowadays,

the application of social media for both collection and dissemination of flood information is increasingly recognised and thus provide an important basis for flood inundation estimation (Fohringer et al., 2015; Smith et al., 2017). The flood related information provided by general public through social media such as Twitter or Weibo, is also effective and valuable calibration-validation data source as an addition to the hardly available traditional monitoring data. Besides, studies (e.g., Joanne et al., 2018; Ziliani et al., 2019) have verified that the combination of data assimilation and numerical model is used

operationally to improve model performance and reduce uncertainties in flood prediction. Ziliani et al. (2019) assimilated field data into the flood model and the prediction result was improved up to 90%. Among the relevant literatures, this method is mostly applied to fluvial flood with the support of satellite-based data or field water level measurements but rarely applied to urban pluvial flood. Moreover, as it is recognised that current two-dimensional (2D) hydrodynamic models are still computationally demanding and challenging for real-time applications at large-scale, recent innovative modelling exploration

has focused on machine learning approach for fluvial flooding, e.g., a deep convolutional neural network (CNN) method has been developed by Kabir et al., (2020). Such an approach is presented for rapid prediction of fluvial flood inundation. However, a good model training still requires good quality inundation data and/or robust hydrodynamic model. Application of machine learning approach in urban flood modelling is promising but still very challenging.

### 5.3 Inter-model and interdisciplinary approaches

Inter-model and interdisciplinary approaches can help to develop the strengths of the various approaches while avoiding shortcomings. Facing the knowledge gap among urban flood risk management, innovative use of computer-based visualization and Virtual Reality (VR) technology has been shown to encourage greater engagement amongst diverse participants. A combined simulation-visualization platform can become an important shared learning tool and there are good prospects for developing an interactive model through the use of computer-based visualisation and virtual reality technology (Wang et al.,

2019; Zhi et al., 2020; Yang et al., 2021). The innovation will be helpful for practitioners to communicate and perceive an extreme flood event. With the help of interactive 3D visualization tools, the extent of inundation and other features such as water depth and floodwater velocity can be better viewed and understood. A combined simulation-visualization approach can enhance decision support by incorporating 2D inundation modelling and 3D data visualization. Besides, as mentioned in the previous part, multi-source data such as social media, remote sensing provides an excellent source of model calibration-

validation data during and after flood events. Its application may be further enhanced when coupled with accelerated real-time urban flood modelling. In other words, the combination between social media data and efficient simulation model provides a strong support to build a real-time surface water flood warning system. Astute combination of models is promising and successfully developed and applied in the future.

### 6 Conclusion

This paper presents a comprehensive review of the current advanced urban flood models. Over the last several decades, there have been a variety of methods based on different theories to model various components of urban flooding, yielding a wide choice space for researchers and practitioners. Based on the discussion above, this paper provides insights into urban flood models, current advances and future challenges. In summary, a drainage network model is a valuable tool with which an urban hydrological method can be coupled. This model is suitable for evaluating and designing a drainage system and assessing flood

risk. SWEs-based models have received much attention in the last ten years because of their capability of reproducing the flow dynamics of urban surface water flooding. Simplified 2D SWEs-based models are widely used in large-scale urban flood simulation with major assumptions for the purpose of regional-/city-/continental-scale risk assessment owing to relatively low computational cost. Full 2D SWEs-based models have proven to be capable of simulating flooding in urban areas with complex urban features, but the particularities of urban areas still pose great challenges in both appropriate model generalisation and

robust numerical algorithms development. Ongoing research on acceleration methods shows promising developments to speed up 2D models and raises hope for real-time applications with better resolutions in the near future. However, reliable modelling of urban surface water flooding will continue to require quality real-world data.

## Author Contributions

K.G. conceptualisation, manuscript writing; M.G. supervision, conceptualisation, manuscript writing, funding acquisition; D.Y. review, editing

## Competing interests

The authors declare that they have no conflict of interest.

## Acknowledgements

The study is financially supported by Early Career Scheme from Hong Kong Research Grant Council (Grant number: 27202419).

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
