# Peer review of "Urban surface water flood modelling – a comprehensive review of current models and future challenges"

_Hydrology and Earth System Sciences, 2020_

## Referee Comment (RC1) · Anonymous Referee #1 · 4 Jan 2021

Dear Editors, This manuscript is a review of existing numerical models to assess the performance of existing and new sewer systems during flooding conditions with surcharging manholes and shallow water on urban surfaces. The limitations and advantages of existing models are highlighted and could become references for the readers in this field, however, there are specific recent publications not considered that are extremely relevant and should be considered. The introduction nicely present the topic discussed and undertaken within the manuscript, however, crucial recent studies conducted within the same field and topic have been completely omitted by the authors. The introduction should be revised and these articles (see below) should be included within the text. The reference, which presents recent modelling issues in the

UK and China, should be included within paragraph 35 - M. Rubinato, A. Nichols, Y. Peng, J. Zhang, C. Lashford, Y. Cai, P. Lin, S. Tait. Urban and river flooding: Comparison of flood risk management approaches in the UK and China and an assessment of future knowledge needs. Water Science and Engineering, 12 (4), 274-283, https://doi.org/10.1016/j.wse.2019.12.004 In paragraph 40, mentioning data availability, it is not clear which data are needed. Nowadays one of the issues is that there is a paucity of data and those available to verify the performance of numerical models are water levels obtained from CCTV cameras, with poor quality. Additionally, in paragraph 50, authors mention the need of high quality data input. Therefore there is the need for experimental/field data to calibrate these models such as those provided by: Md.N.A. Beg, R.F. Carvalho, S. Tait, W. Brevis, M. Rubinato, A. Schellart, J. Leandro. (2018) A comparative study of manhole hydraulics using stereoscopic PIV and different RANS models. Water Science and Technology, 2017 (1), 87-98, http://wst.iwaponline.com/content/early/2018/02/28/wst.2018.089 Rubinato, J. Shucksmith, A. J. Saul, W. Shepherd. (2013) Comparison between Infoworks results and a physical model of an urban drainage system, Water Science and Technology, 68 (2), 372-379, http://wst.iwaponline.com/content/68/2/372 Gómez, M., and B. Russo. 2011. "Methodology to Estimate Hydraulic Efficiency of Drain Inlets." Proceedings of the Institution of Civil Engineers - Water Management 164 (2): 81-90. doi:10.1680/ wama.900070 Lopes, P., J. Leandro, R. F. Carvalho, P. Páscoa, and R. Martins. 2015. "Numerical and Experimental Investigation of a Gully under Surcharge Conditions." Urban Water Journal 12 (6): 468-476. doi:10.1080/ 1573062X.2013.831916 Gómez, M.; Russo, B.; Tellez-Alvarez, J. Experimental investigation to estimate the discharge coefficient of a grate inlet under surcharge conditions. Urban Water J. 2019, 16, 85-91. M. Rubinato, R. Martins, J. Shucksmith. (2018) Quantification of energy losses at a surcharging manhole. Urban Water Journal, 15 (3), 234-241, https://www.tandfonline.com/doi/full/10.1080/1573062X.2018.1424217 Tellez-Alvarez, J., Gomez, M., Russo, B. (2020) Quantification of energy loss in two grated inlets under pressure. Water, 12, 1601, doi:10.3390/w12061601 M. Rubinato, L. Seungsoo,

R. Martins, J. Shucksmith. (2018) Surface to sewer flow exchange through circular inlets during urban flood conditions. Journal of Hydroinformatics, 20 (3), 564-576, DOI: 10.2166/hydro.2018.127 . R. Martins, M. Rubinato, G. Kesserwani, J. Leandro, S. Djordjevic, J. Shucksmith. (2018) On the characteristics of velocity fields on the vicinity of manhole inlet grates during flood events. Water Resources Research, 54 (9), 6408-6422, https://doi.org/10.1029/2018WR022782 Lee, S., H. Nakagawa, K. Kawaike, and H. Zhang. 2015. "Urban Inundation Simulation considering Road Network and Building Configurations." Journal of Flood Risk Management 9 (3): 224-233. doi:10.1111/ jfr3.12165 Furthermore, during flooding events, there is the health risk associated with polluted water reaching areas where humans live and there is the need not only to characterize the hydraulic aspect of flooding events, but the pollutant transport too. Please see for example the paper below: MNA Beg, M Rubinato, RF Carvalho, JD Shucksmith. CFD Modelling of the Transport of Soluble Pollutants from Sewer Networks to Surface Flows during Urban Flood Events. Water 12 (9), 2514 METHODOLOGY The initial part composed by paragraphs 65-80 is not using a scientific language and should be revised to avoid expressions such as "this method can help scholars...", "this method was conducted in ... and Google Scholar", it is normal that the research should be conducted via libraries and Google Scholar and these details are not needed. When authors introduce the equations they completely miss the crucial aspect of linking the two systems, pipes and urban streets. The weir and orifice equations are extremely important to quantify the flow exchange between these two systems and therefore please have a look at the following manuscript: M. Rubinato, R. Martins, G. Kesserwani, J. Leandro, S. Djordjevic, J. Shucksmith. (2017) Experimental calibration and validation of sewer/surface flow exchange equations in steady and unsteady flow conditions. Journal of Hydrology, 552, 421-432, https://doi.org/10.1016/j.jhydrol.2017.06.024 M. The manuscript can be definitely be considered for publication once the minor final comments are addressed. Authors have provided a rich analysis of existing techniques and limitations, however the lack of significant relevant studies needs to be dealt with before publication in this manuscript.

СЗ

---

## Referee Comment (RC2) · Anonymous Referee #2 · 5 Jan 2021

This paper shows a review of urban flood models for inundation prediction. The structure was clearly organized and presented. But I expect the detailed model description and comparisons which can derive specific conclusions and provide informative insights for model users. This paper still needs to be largely modified in terms of model comparisons and English language.

1.The introduction emphasized the importance of urban flooding and explained the reasons to choose this topic. I doubt that there was less emphasis on urban surface water flooding compares to fluvial and coastal flooding. Please specify this. How does urban flooding will increase in severity and frequency due to climate change in the

future? 2.every figure and table needs to be explained in details in the text. 3. It is suggested to explain the physics behind the model clearly. In the section 3, please explain the equations clearly. 4. it is not suggested to use words like 'some' and 'very' and in the text. 5. please explain horizontal coupling and vertical coupling (in Figure 5) in the text. Figure 6 is not clear. What does green area mean in Figure 6? 6. Line 310: CA models can produce reliable results..... but the authors commented that CA approaches ..... still need to be further verified in line 315. 7. The future challenges of these models needs to be discussed thoroughly.

Please also note the supplement to this comment:
https://hess.copernicus.org/preprints/hess-2020-655/hess-2020-655-RC2-supplement.pdf

**Supplement:**

[revised manuscript text omitted]

---

## Author Comment (AC1) · 9 Mar 2021

Dear Editors, this manuscript is a review of existing numerical models to assess the performance of existing and new sewer systems during flooding conditions with surcharging manholes and shallow water on urban surfaces. The limitations and advantages of existing models are highlighted and could become references for the readers in this field, however, there are specific recent publications not considered that are extremely relevant and should be considered. The introduction nicely presents the topic discussed and undertaken within the manuscript, however, crucial recent studies conducted within the same field and topic have been completely omitted by the authors.

**Response**: Thanks for the comments. We have carefully re-sorted out relevant literatures suggested by the reviewer and revised our manuscript accordingly. Below are our response point by point.

The introduction should be revised and these articles (see below) should be included within the text. The reference, which presents recent modelling issues in the UK and China, should be included within **paragraph 35** - M. Rubinato, A. Nichols, Y. Peng, J. Zhang, C. Lashford, Y. Cai, P. Lin, S. Tait. **Urban and river flooding: Comparison of flood risk management approaches in the UK and China and an assessment of future knowledge needs**. Water Science and Engineering, 12 (4), 274-283.

**Response**: Thank you for suggestion. The reference is added in the paragraph 35 in the revised manuscript.

In **paragraph 40**, mentioning data availability, it is not clear which data are needed. Nowadays one of the issues is that there is a paucity of data and those available to verify the performance of numerical models are water levels obtained from CCTV cameras, with poor quality. Additionally, in paragraph 50, authors mention the need of high-quality data input. Therefore, there is the need for **experimental/field data to calibrate these models** such as those provided by:

- Md.N.A. Beg, R.F. Carvalho, S. Tait, W. Brevis, M. Rubinato, A. Schellart, J. Leandro. **A comparative study of manhole hydraulics using stereoscopic PIV and different RANS models**. Water Science and Technology, 2017 (1), 87-98.
- Rubinato, J. Shucksmith, A. J. Saul, W. Shepherd. (2013) **Comparison between Infoworks results and a physical model of an urban drainage system**, Water Science and Technology, 68 (2), 372–379.
- Gómez, M., and B. Russo. 2011. "**Methodology to Estimate Hydraulic Efficiency of Drain Inlets**." Proceedings of the Institution of Civil Engineers - Water Management 164 (2): 81–90.
- Lopes, P., J. Leandro, R. F. Carvalho, P. Páscoa, and R. Martins. 2015. "**Numerical and Experimental Investigation of a Gully under Surcharge Conditions**." Urban Water Journal 12 (6): 468–476.
- Gómez, M.; Russo, B.; Tellez-Alvarez, J. **Experimental investigation to estimate the discharge coefficient of a grate inlet under surcharge conditions**. Urban Water J. 2019, 16, 85–91.
- M. Rubinato, R. Martins, J. Shucksmith. (2018) **Quantification of energy losses at a surcharging manhole**. Urban Water Journal, 15 (3), 234-241.

- Tellez-Alvarez, J., Gomez, M., Russo, B. (2020) **Quantification of energy loss in two grated inlets under pressure**. Water, 12, 1601.
- M. Rubinato, L. Seungsoo, R. Martins, J. Shucksmith. (2018) **Surface to sewer flow exchange through circular inlets during urban flood conditions**. Journal of Hydroinformatics, 20 (3), 564–576.
- R. Martins, M. Rubinato, G. Kesserwani, J. Leandro, S. Djordjevic, J. Shucksmith. (2018) **On the characteristics of velocity fields on the vicinity of manhole inlet grates during flood events**. Water Resources Research, 54 (9), 6408- 6422.
- Lee, S., H. Nakagawa, K. Kawaike, and H. Zhang. 2015. "**Urban Inundation Simulation considering Road Network and Building Configurations**." Journal of Flood Risk Management 9 (3): 224–233.

**Response**: We appreciate the reviewer for his additional suggestion on relevant literatures. Some of the representative papers (Rubinato et al.,2013; Lopes et al., 2015; Gómez et al., 2019; Martins et al., 2018) have been reviewed and added in the Section 4.2 to clarify the importance of experimental/field data in model development. Some other relevant papers (Rubinato et al.,2017&2018) are added to the Section 3.2.3 to support our review in terms of quantification of drainage flows.

Furthermore, during flooding events, there is the health risk associated with polluted water reaching areas where humans live and there is the need not only to characterize the hydraulic aspect of flooding events, but the pollutant transport too. Please see for example the paper below: MNA Beg, M Rubinato, RF Carvalho, JD Shucksmith. CFD Modelling of the Transport of Soluble Pollutants from Sewer Networks to Surface Flows during Urban Flood Events. Water 12 (9), 2514

**Response**: We agree with the reviewer. The health risk associated with pollutant transport during urban flooding is indeed an important issue to be modelled and overcome. Urban flooding can cause a surcharge of sewer flow, flush pollutants and wastewater to public area, so causing health risks for the people, such as breakout of epidemic disease, and drinking water pollution (Beg et al., 2020). However, surface water pollution heavily relies on the surface water dynamics. As this manuscript only focusses on the flood dynamic simulation, we did not go through in-depth review and discussion on pollutant transport associated with surface water dynamics. To avoid the ambiguity, we have clarified the aim of the review paper in Section 1.

METHODOLOGY The initial part composed by paragraphs 65-80 is not using a scientific language and should be revised to avoid expressions such as "this method can help scholars. . .", "this method was conducted in . . . and Google Scholar", it is normal that the research should be conducted via libraries and Google Scholar and these details are not needed.

**Response**: Thanks for the suggestion. We have rephased relevant sentences to be more scientific expression.

When authors introduce the equations they completely miss the crucial aspect of linking the two systems, pipes and urban streets. The weir and orifice equations are extremely important to quantify the flow exchange between these two systems and therefore please have a look at the following manuscript: M. Rubinato, R. Martins, G. Kesserwani, J. Leandro, S. Djordjevic, J. Shucksmith. (2017) Experimental calibration and validation of sewer/surface flow exchange equations in steady and unsteady flow conditions. Journal of Hydrology, 552, 421-432.

**Response**: We have reviewed the suggested paper in our previous version. This has been re-mentioned in Section 3.2.3 when we discuss the approaches to quantify the exchange of surface water and pipe flow.

The manuscript can be definitely be considered for publication once the minor final comments are addressed. Authors have provided a rich analysis of existing techniques and limitations; however, the lack of significant relevant studies needs to be dealt with before publication in this manuscript.

**Response**: We appreciate the reviewer for his/her positive comments. We have addressed the comments point by point and revised our manuscript accordingly following your suggestions.

---

## Author Comment (AC2) · 9 Mar 2021

**Response to Reviewer 2**

**Anonymous Referee #2**

This paper shows a review of urban flood models for inundation prediction. The structure was clearly organized and presented. But I expect the detailed model description and comparisons which can derive specific conclusions and provide informative insights for model users. This paper still needs to be largely modified in terms of model comparisons and English language.

**Response**: We appreciate the reviewer for the suggestions. As shown in Table 3, there are many representative urban flood models in research community, so we believe it is not feasible to describe these models individually one by one. However, we do classify these models into four categories based on the model structures and assumptions. As described in each sub-section of Section 3, the model conceptualization, governing equations and key features have been discussed type by type. Following the reviewer's suggestion, we have done further comparison between the four types of models in Section 4 and summarized the key comparative information and the pros and cons of each type of model in Table 4. Moreover, our manuscript has been proof-read by professional English native editor from Cambridge Editing Service, a proof document is shown below.

[Figure]

Order No. 944-13-26

**Editor's Report**

Thank you for the opportunity to edit your paper. It was a pleasure to review your work, which was engaging. I have focused on correcting the grammar and improving sentence structure throughout and have applied UK English conventions, as requested.

Please carefully read through my edits and in-text comments—which provide further detail and suggestions for improvements—before accepting or rejecting any changes. In addition, please review the table below for an assessment of your manuscript and an overview of key points that have been addressed.

I wish you the best of luck with your paper, and I look forward to working with you again soon.

Sincerely,

Kimberly S, PE

| Summary | |
|---|---|
| Content | Your review of urban surface water modelling was presented clearly and logically. |
| Grammar and punctuation | Your grammar and punctuation were very good with the except of a few missed items. |
| Tone and register | The language was appropriate. I only had to make a few suggestions for more formal terms. |
| Style | There are quite a few long paragraphs and long sentences than can benefit from being split in two. Additionally, there were overused words, e.g. "widely", "reasonable", and "decades". |
| Format | There were no major issues with formatting. I ensured each heading stayed with the paragraph it preceded (i.e. "keep with next"). |

1.The introduction emphasized the importance of urban flooding and explained the reasons to choose this topic. I doubt that there was less emphasis on urban surface water flooding compares to fluvial and coastal flooding. Please specify this. How does urban flooding will increase in severity and frequency due to climate change in the future?

**Response:** Thanks for pointing this out. What we mean is that **historically** coastal and fluvial flooding has been paid more significant attention compared to pluvial flooding, e.g., we have found from literatures that a lot of flood models have been initially developed for the purpose of fluvial and coastal flood modelling more than 15 years ago, but less on urban surface water flooding. For example, the Pitt Review by Cabinet Office UK (2008) commented that during floods that affected the UK in the summer of 2007, two thirds of the damage in urban areas was caused by surface water flooding, for which *no models, forecasts, warnings or management strategies* existed. Since then, surface water flooding in urban areas due to intense rainfall has increasingly attracted attentions in recent decade.

Regarding the increasing in severity and frequency due to climate change in the future, we believe that a lot of scientific papers and reports (e.g., UNISDR, 2015; IPCC, 2013; Bernet et al., 2017; Barredo, 2009; Zhou et al., 2013; Moncoulon et al., 2016) have indicated that urban surface water flooding is expected to increase in severity and frequency in the future with urbanization, economic development, and more frequent extreme weather. For example, the United Nations (UN) recently reported that the world's urban population is projected to grow both in absolute terms, and as a fraction of a growing global population. As more people move to cities, they inevitably turn green areas into impervious areas, increasing urban surface runoff. And as more people and properties in urban areas, flood severity would be definitely increased. Synthesis Report: Climate Change 2014 by IPCC also reported that "*extreme precipitation events will become more intense and frequent in many regions*" and "*recent detection of increasing trends in extreme precipitation and discharges in some catchments implies greater risks of flooding on a regional scale*". The increasing of frequency arises from the possibility for climate change to lead to more extreme rainfall, which is the main reason leading to the urban surface flooding.

*United Nations Department of Economic and Social Affairs/Population Division, 2012. World Urbanization Prospects: The 2011 Revision. New York: United Nations.*
*IPCC Fifth Assessment Report (AR5)(Cambridge Univ. Press, 2014)*

We have specified this in the introduction of the revised manuscript.

2. every figure and table need to be explained in details in the text.

**Response:** Following the reviewer's suggestion, we have provided detailed description to each figure and table in the revised manuscript text (please see the tracked changes).

3. It is suggested to explain the physics behind the model clearly. In the section 3, please explain the equations clearly.

**Response:** Following the reviewer's suggestion, we have discussed more on the model equations and the physics behind the model. Please see the tracked changes in page 7&8 of the revised manuscript.

4. it is not suggested to use words like 'some' and 'very' and in the text.

**Response:** Following the reviewer's suggestion, we have revised the text.

5. please explain horizontal coupling and vertical coupling (in Figure5) in the text. Figure 6 is not clear. What does green area mean in Figure 6?

**Response:** For figure 5 and 6, the more detailed information is described as below,

*The horizontal coupling means components exchanging between the 2D surface cell and the drainage cell in horizontal direction. And vertical coupling means the discharge redistribution in the drainage cell, to calculate surcharge from the urban surface into the drainage or backflow from the pipe.*

*The green area in the Figure 6 is the boundary area, where bidirectional components exchanging between hydraulic and hydrological areas is allowed to update the calculation.*

6. Line310: CA models can produce reliable results..... but the authors commented that CA approaches ..... still need to be further verified in line 315.

**Response:** Based on the published paper, the analysis of the numerical cases showed that CA models can be valuable tools for flood simulation. However, as an emerging method, there is only limited literature and most of the tests are ideal cases. Besides, it showed less accuracy in reproducing two-dimensional flow dynamics in respect to a model based on the full shallow water equations. CA models, with high computational performance and acceptable compromise in accuracy, are more suitable for large domains or a significantly large number of simulations. When only few simulations are performed, they are not the most suitable choice (Dottori & Todini, 2011; Guidolin et al., 2016). So, in this concern, we stated that CA approach still needs to be further verified in 1D and 2D practical applications.

**Response:** The future challenges section has been re-organized as below. This has been updated in the revised manuscript.

**5.1 Refinement of SWEs-based models**

[revised manuscript text omitted]